# Evolution and Multi-Scenario Prediction of Land Use and Carbon Storage in Jiangxi Province

Yue Huang [1], Fangting Xie [2], Zhenjiang Song [1] and Shubin Zhu [1],*

1. School of Economics and Management, Jiangxi Agricultural University, Nanchang 330100, China; huangyue@stu.jxau.edu.cn (Y.H.); tgsongzhenjiang@126.com (Z.S.)
2. Research Academy for Rural Revitalization of Zhejiang Province, Zhejiang A&F University, Hangzhou 311300, China; xiefangting@zafu.edu.cn
* Correspondence: shubinzhu@163.com

**Abstract:** In recent years, escalating global warming and frequent extreme weather events have caused carbon emission reduction to become a pressing issue on a global scale. Land use change significantly impacts ecosystem carbon storage and is a crucial factor to consider. This study aimed to examine the evolutions in land use and their impact on carbon storage in Jiangxi Province, China. Using the coupled PLUS-InVEST model, we analyzed the spatial patterns alterations of both land use and carbon storage from 2000 to 2020 and set four scenarios for 2040. Our findings indicated the following: (1) From 2000 to 2020, the area of cropland, forest, grassland, and unused land declined, whereas the area of water and built-up land increased, with changes mainly occurring in 2010–2020. (2) From 2000 to 2020, due to the land use change, carbon storage in Jiangxi Province demonstrated a decreasing trend, with a total reduction of $2882.99 \times 10^4$ t. (3) By 2040, under the dual protection scenario for cropland and ecology, the expansion of built-up land will be most restricted among the four scenarios, and the largest projected carbon storage was foreseen. This suggests that carbon loss can be minimized by focusing on cropland and ecological conservation, especially forests. Our research findings can facilitate policy decisions to balance economic development and environmental protection in Jiangxi Province in the future.

**Keywords:** land use; carbon storage; PLUS-InVEST model; multi-scenario simulation





## 1. Introduction

Terrestrial ecosystems, recognized as the largest carbon reservoirs, can effectively sequester atmospheric $CO_2$ [1]. They are crucial to the global carbon cycle and climate change mitigation [2,3]. Notably, carbon storage varies substantially across various land use types [4,5]. The ability of surface plants and soils to store carbon fluctuates with land use changes, significantly influencing the carbon storage of terrestrial ecosystems [6,7]. Over the past century, rapid industrialization and land use changes driven by human activities have critically disturbed the carbon equilibrium of terrestrial ecosystems. This disruption has led to climate-related and environmental problems in nearly all cities worldwide, such as the greenhouse effect and extreme weather events [8]. Notably, over the past 150 years, one-third of $CO_2$ emissions have been directly attributed to artificial land use changes [9].

The impact of land use and cover changes on the terrestrial carbon cycle has emerged as a vital aspect of climate change research [10,11]. Research indicates that rapid urban expansion has significantly reduced ecological land, with cropland being the primary type encroached upon by urbanization [12]. This change has resulted in substantial loss of carbon storage [13,14]. Once this transformation occurs, short-term restoration of carbon storage to its initial level is challenging [15]. A study by Li et al. [16] revealed that due to urbanization, the net primary productivity (NPP) loss in China from the late 1980s to 2015 reached 1.695 Tg C, with 63.02% resulting from the conversion of cropland to built-up land.

Currently, carbon storage evaluation primarily employs methods such as field investigation, simulation models, and remote sensing inversion [17], in which the InVEST model can identify the relationship between the changes in land use and carbon storage. It has been commonly used due to its benefits of requiring fewer parameters, simple operation, fast execution speed, and visualization of estimation results [18,19]. There are several land use simulation models for simulating the spatial distribution of land use. Widely used models include the CA-Markov model [20,21], ANN-CA model [22,23], CLUE-S model [24,25], and FLUS model [26,27]. However, these models are ineffective in identifying the factors influencing land use and do not allow for the simulation of multiple land use patches [28]. The PLUS model employs the adaptive inertial competition mechanism, which allows land use types to gradually adjust their distribution based on the current state and the surrounding land use types. In addition, the PLUS model utilizes the roulette competition mechanisms that introduces a random element to the competition process Moreover, the PLUS model goes beyond the traditional CA model by combining future predicted variables to calculate the development potential of each land use type, which include factors such as economic development, population growth, and environmental conditions. Through the random forest algorithm, the PLUS model can better capture the dynamics of land use changes and provide more realistic and accurate simulations [29,30]. The coupled InVEST-PLUS model was used in this study based on the advantages of the InVEST model and PLUS model, as well as the fact that they are both based on the raster data of land use, which can effectively link between land use and carbon storage. The coupled InVEST-PLUS model has been widely used—for example, scholars such as Li et al. [8] and Wang et al. [31] have used the PLUS-InVEST coupled model to analyze the evolution of land use and carbon storage in Kunming and GBA, respectively. However, studies on a provincial scale are sparse, which is not conducive to comprehensive integrated planning at the province level.

Boasting superior geographical and climatic conditions, Jiangxi Province is abundant in forest resources, with the second-highest forest coverage rate in China at 61.16%. Additionally, Poyang Lake is situated in the north of Jiangxi Province, which is China's largest freshwater lake. In July 2014, Jiangxi Province was selected as one of the first national demonstration zones for ecological civilization. However, with the province's rapid economic and societal development, conflicts between ecological protection and economic development are becoming more apparent. Therefore, in the context of sustainable development, it is crucial to analyze the evolution and scenario prediction in Jiangxi Province.

Utilizing the coupled PLUS-InVEST model, this study delved into the evolution of land use and carbon storage in Jiangxi Province from a spatial and quantitative perspective for the period of 2000–2020. In order to explore the land use change over the next two decades under different development scenarios in Jiangxi Province and produce rational land use planning, this study simulated the land use patterns under four scenarios in 2040, considering the impacts of terrain and soil factors, climatic conditions, social factors, transportation elements, and water systems. The aims of this study were as follows: (1) investigate the characteristics of land use change in Jiangxi Province from 2000 to 2020 and its impact on carbon storage; (2) simulate and analyze the characteristics of land use and carbon storage in 2040 under four scenarios; (3) identify land use conflicts arising during the course of social development and provide suggestions for future land use development planning in Jiangxi Province. The findings of this study can identify patterns of land use change in the development of Jiangxi province and provide a policy reference for alleviating the contradiction between economic development and environmental protection in Jiangxi Province in the future.

## 2. Study Area and Data

### 2.1. Study Area

Jiangxi Province, situated on the southern bank at the juncture of the middle and lower reaches of the Yangtze River, plays a crucial role of the Yangtze River Economic Belt. As depicted in Figure 1, Jiangxi Province is an inland province, located in the southeast of China (24°29′–30°04′ N, 113°34′–118°28′ E), comprising 11 cities and spanning an area of 167,000 km². By the end of 2021, the population of Jiangxi Province exceeded 45.174 million, of which the urban population accounted for 61.46%. Jiangxi Province enjoys a subtropical and temperate climate, characterized by warmth, ample sunlight, and abundant precipitation year-round. The topography of Jiangxi Province is predominantly mountainous and hilly, with the terrain progressing from the periphery to the interior and from south to north, shaping a vast basin opening to the north.

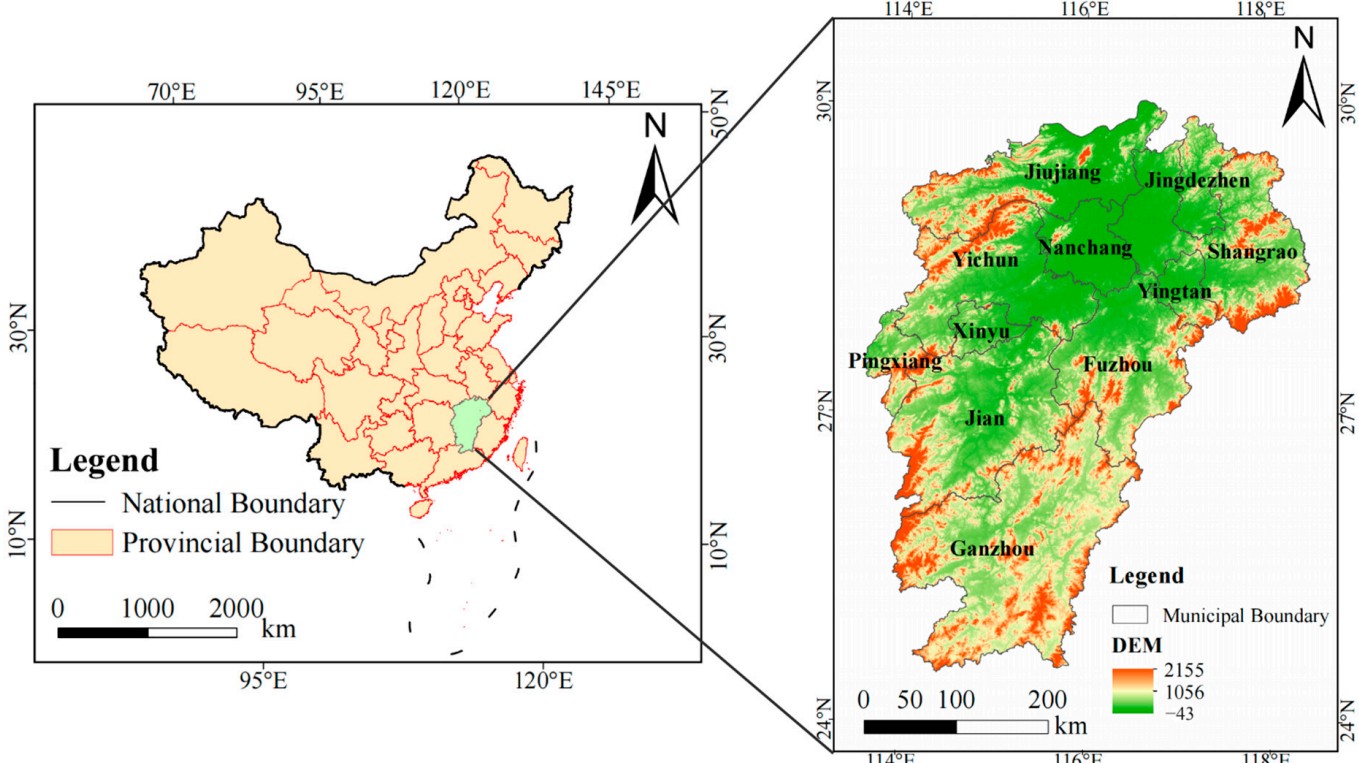

**Figure 1.** Geographical location and topography of Jiangxi Province.

### 2.2. Data Source

This paper utilized land use raster data in 2000, 2010, and 2020, as well as 13 driving factor data points that were employed for land use simulation projections. The land use data, which have a spatial resolution of 30 m, were sourced from the Resource and Environment Science Data Center (https://www.resdc.cn/, accessed on 9 April 2023). In line with the National Land Use/Cover Classification System for Remote Sensing Monitoring and the specific conditions of the study area, this paper classified land use types into six primary classifications: cropland, forest, grassland, waters, built-up land, and unused land. The expansion of land use is influenced by both natural factors and social factors. Drawing from previous studies [32,33], 13 driving factors were chosen for this paper, including terrain and soil factors, meteorological factors, social factors, traffic factors, and water system factors. The details are presented in Table 1. All raster data were unified into Xian 1980 coordinate system and resampled to a 30 m resolution through ArcGIS 10.0.

**Table 1.** Data information of driving factors.

| Data Name | Data Accuracy | Data Source |
|---|---|---|
| DEM | 30 m | Data Center for Resource and Environmental Sciences, Chinese Academy of Sciences (https://www.resdc.cn/, accessed on 9 April 2023) |
| Slope | | Obtained by DEM extraction |
| Average annual temperature | | Data Center for Resource and Environmental Sciences, Chinese Academy of Sciences (https://www.resdc.cn/, accessed on 9 April 2023) |
| Average annual precipitation | | |
| GDP | | |
| Population | | |
| Distance to urban primary roads | 1 km | |
| Distance to urban secondary roads | | |
| Distance to urban tertiary roads | | Open Street Map (https://www.openstreetmap.org/, accessed on 9 April 2023) |
| Distance to highway | | |
| Distance to railroad | | |
| Distance to water system | | |

## 3. Research Methods

This study was divided into two parts. The first part analyzed the evolution of land use and carbon storage from 2000 to 2020, where carbon storage was calculated by the InVEST model based on land use data. And the conversions of land use types were obtained through transfer matrix of land use. The second part predicted 4 scenarios of land use through the PLUS model, and similarly, the carbon storage under the 4 scenarios was calculated through the InVEST model. The framework of this study is shown in Figure 2.

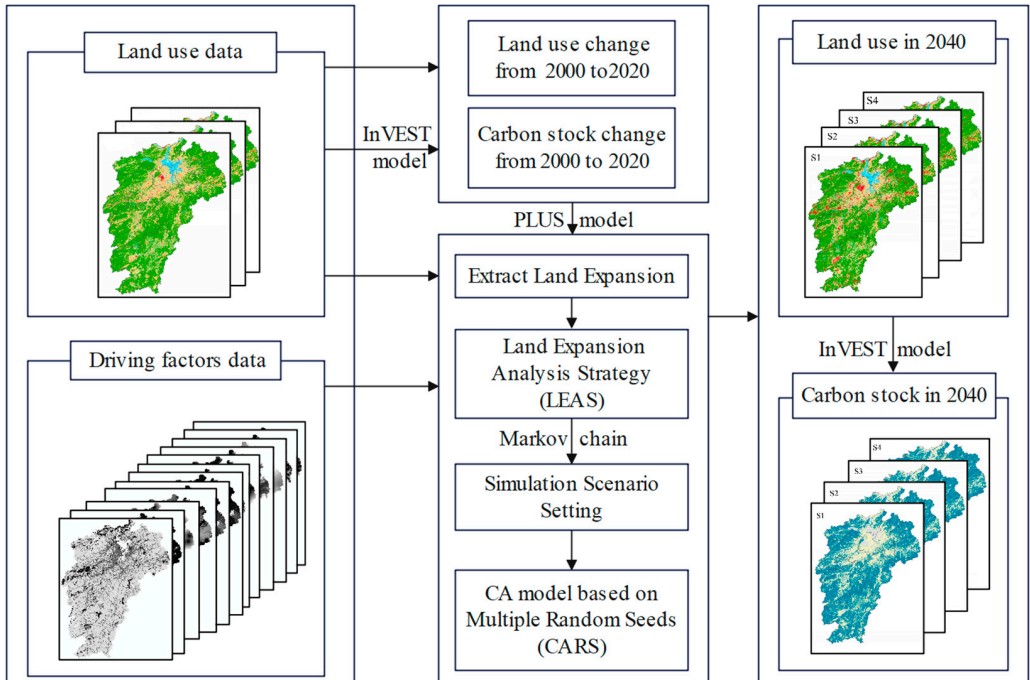

**Figure 2.** Research framework.

### 3.1. Transfer Matrix of Land Use

The transfer matrix of land use is a frequently used approach for analyzing land use change, essentially serving as a key aspect of land use/cover research [34]. Essentially, the land use transfer matrix applies the Markov model to examine land use change. It is a primary method for quantitatively studying the quantity and directionality of mutual transformations between different land use types. Furthermore, it can accurately depict the structural properties of land use change and the direction of transfer among different land use types [35]. The formula is as follows:

$$K_{ab} = \begin{vmatrix} k_{11} & k_{12} & \cdots & k_{1n} \\ k_{21} & k_{22} & \cdots & k_{2n} \\ \vdots & \vdots & & \vdots \\ k_{n1} & k_{n2} & \cdots & k_{nn} \end{vmatrix} \quad (1)$$

In Formula (1), $K$ stands for the variation area of each land use type; $a$ and $b$ denote the start and finish of the study period, respectively; and $n$ is the number of land use types.

The land use type transfer-in/transfer-out contribution ratio can reveal the characteristics of land use change. The formulas are as follows:

$$S_{i+} = \frac{\sum_{j=1}^{n} S_{ji}}{S_t} \times 100\% \quad (2)$$

$$S_{i-} = \frac{\sum_{j=1}^{n} S_{ji}}{S_t} \times 100\% \quad (3)$$

In Formula (2), $S_{i+}$ represents the transfer-in contribution of land use type $i$, which is the proportion of the area transferred from other land use types to land use type $i$ in relation to the total area of land transferred. Conversely, in Formula (3), $S_{i-}$ denotes the transfer-out contribution of land use type $i$, which is the proportion of the area transferred from land use type $i$ to other land use types relative to the total area of land transferred. $S_t$ stands for the total land transfer area, $S_{ji}$ represents the transfer from land use type $j$ to $i$, and n is the number of land use types [36].

### 3.2. InVEST Model

The Carbon Storage and Sequestration module of the InVEST 13.2.1 (Integrated Valuation of Ecosystem Services and Tradeoffs) software was utilized to calculate the carbon storage of Jiangxi Province. The model has been widely used due to its ability to geographically integrate changes in land use and dynamics of terrestrial ecosystem carbon storage while effectively articulating the relationships between such changes [14,30]. The InVEST model categorizes carbon storage into four fundamental carbon pools: above-ground biomass carbon, below-ground biomass carbon, soil organic carbon, and dead organic matter carbon [37]. The calculation formula is as follows:

$$C_{tot} = \sum_{i=1}^{6} \left( C_{above\_i} + C_{below\_i} + C_{soil\_i} + C_{dead\_i} \right) \times A_i \quad (4)$$

In Formula (4), $C_{tot}$ represents the total carbon storage; $i$ is the number of land use types; and $C_{above\_i}$, $C_{below\_i}$, $C_{soil\_i}$, and $C_{dead\_i}$ denote the carbon storage of above-ground biomass, below-ground biomass, soil, and dead organic matter for land use type $i$, respectively. $A_i$ stands for the area of land use type $i$. The carbon density of each land type was referenced from literature data [38], as shown in Table 2.

**Table 2.** Carbon density of each land use type (Mg C/hm$^2$).

| Land Use Type | $C_{above}$ | $C_{below}$ | $C_{soil}$ | $C_{dead}$ |
|---|---|---|---|---|
| Cropland | 3.55 | 2.09 | 32.34 | 0.54 |
| Forest | 46.9 | 11.2 | 42.3 | 0.69 |
| Grassland | 1.02 | 8.45 | 52.52 | 0.43 |
| Waters | 0.08 | 0.07 | 0.00 | 0.00 |
| Built-up land | 1.49 | 0.35 | 0.04 | 0.00 |
| Unused land | 0.36 | 0.53 | 1.81 | 0.03 |

### 3.3. PLUS Model

The PLUS 1.4 (Patch-generating Land Use Simulation) software was developed by the High-Performance Spatial Computing Intelligence Laboratory at the China University of Geosciences in Wuhan, combining the Land Expansion Analysis Strategy (LEAS) and the CA model based on Multiple Random Seeds (CARS) [29]. The PLUS model makes use of the random forest algorithm to discern the contribution of each driving factor to various land use types and utilizes the Markov chain or linear regression to simulate the expansion demands for various land use types. The PLUS model stands out from other models and has been widely employed due to its ability to fuse the impacts of multiple spatial elements with the dynamics of geographic units in simulating land use change, thereby achieving greater precision and more realistic landscape patterns [28].

The formulation principle of the PLUS model has been thoroughly described in prior literature [39,40]. Hence, this paper only provides a brief overview of its process: (1) the land use data were converted by PLUS, enabling the PLUS model to identify them, and following this, land expansion areas from 2000 to 2020 were extracted; (2) the LEAS model was used to determine the contribution of driving factors to the change of each land use type; (3) the land demands of each land use type for 2040 were obtained through Markov chain; (4) finally, the parameters and conditions of the CARS model need to be set in order to produce the simulated land use raster data. In the transfer matrix, '1' indicates that a transfer was permissible and '0' indicates that it was not. Moreover, water areas were set as the limiting area.

The parameters of the PLUS model were set as follows: (1) in the LEAS model, the number of regression tree was 20, sampling rate was 0.01, mTry was 13, and the thread was 1; (2) in the CARS model, the neighborhood size was 3, patch generation threshold was 0.5, expansion coefficient was 0.1, percentage of seeds was 0.0001, and the thread was 3; (3) the neighborhood weights were obtained through calculating the proportion of the expanded area of each land use type to the total area, which was obtained through Markov chain. We utilized land use data from 2000 to 2010 to simulate land use in 2020 to check the accuracy of the simulation. As a result, the kappa coefficient was 0.76 at a sampling rate of 0.5, indicating that the simulation results were highly reliable.

### 3.4. Simulation Scenario Setting

In line with the development requirements of the 14th Five-Year (2021–2025) Plan for the Protection and Utilization of Natural Resources in Jiangxi Province, which prioritizes cropland protection, resource conservation, ecological protection, and restoration, as well as adhering to the principles of ecological priority and green development, this paper established four scenarios. These were the natural development scenario (S1), the cropland protection scenario (S2), the ecological protection scenario (S3), and the dual protection scenario for cropland and ecology (S4). The transfer probabilities for each land use type in the Markov chain were adjusted in different scenarios [41].

Under the S1, the evolution of each land use type followed the same pattern observed from 2000 to 2020. There were significant conversions between all land use types, with the exception of the small areas of built-up land and unused land during 2000–2020. Consequently, the conversions between built-up land and unused land were set to 0, while the other conversions were set to 1.

Under the S2, based on S1, conversions of cropland to other land use types were minimized, with the probability of transferring cropland to other types reduced by 50% in comparison to natural development.

Under the S3, the protection of cropland, forest, waters, and grassland was enhanced, while the expansion of built-up land was restrained. Based on S1, the probabilities of transitioning cropland, forest, grassland, and waters to built-up land were reduced by 30%, 50%, 20%, and 20%, respectively. The probability of transitioning forest to grassland was cut by 50%, the probabilities of transitioning waters and grassland to forest were increased by 20%, and the probability of transitioning built-up land to forest was increased by 10%.

Finally, under the S4, cropland settings were determined according to S2, while other land use types were configured following the principles of S3.

## 4. Results and Analysis

### 4.1. Land Use Change from 2000 to 2020

#### 4.1.1. Spatial Distribution and Changes in Land Use

Figure 3 illustrates the spatial distribution of land cover in Jiangxi Province for the years 2000, 2010, and 2020. The province's land cover is primarily composed of forest and cropland, accounting for approximately half and one-third of Jiangxi's total area, respectively. Poyang Lake in the south and its tributaries are the main part of water system in Jiangxi Province. The area of unused land is the smallest, constituting less than 0.2% of the total land area, primarily composed of the beaches exposed by the receding water level of Poyang Lake during drought periods. Furthermore, it is evident that built-up land has been progressively expanding from 2000 to 2020, primarily clustered around Nanchang, which is the capital city of Jiangxi Province, and extending in a strip towards the southwest and northeast.

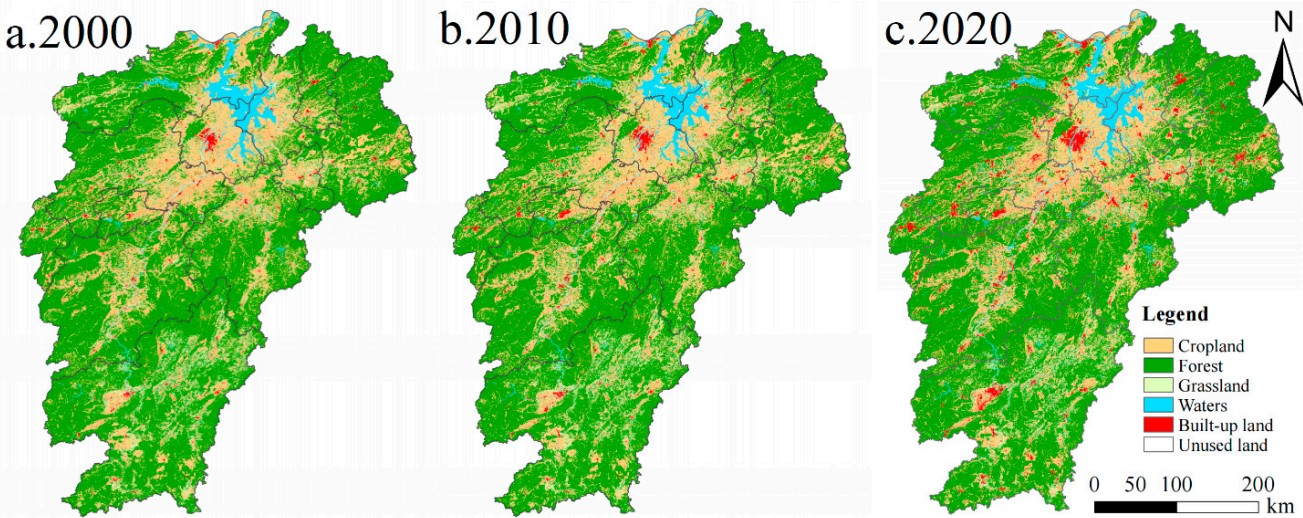

**Figure 3.** Spatial distribution of land use types in Jiangxi Province from 2000 to 2020.

Figure 4 presents the changes in land use from 2000 to 2020. Among all types, built-up land witnessed the most significant expansion, which increased by 3472.10 km$^2$, reflecting the rapid urbanization in Jiangxi Province over the last 20 years. Additionally, the area of waters expanded by 1550.38 km$^2$, indicating strong protection measures for waters. Conversely, the areas of cropland, forest, grassland, and unused land decreased by 2557.59 km$^2$, 1283.93 km$^2$, 1064.25 km$^2$, and 116.71 km$^2$, respectively. Notably, the changes in all land use types from 2010 to 2020 were significantly more pronounced than the changes observed from 2000 to 2010.

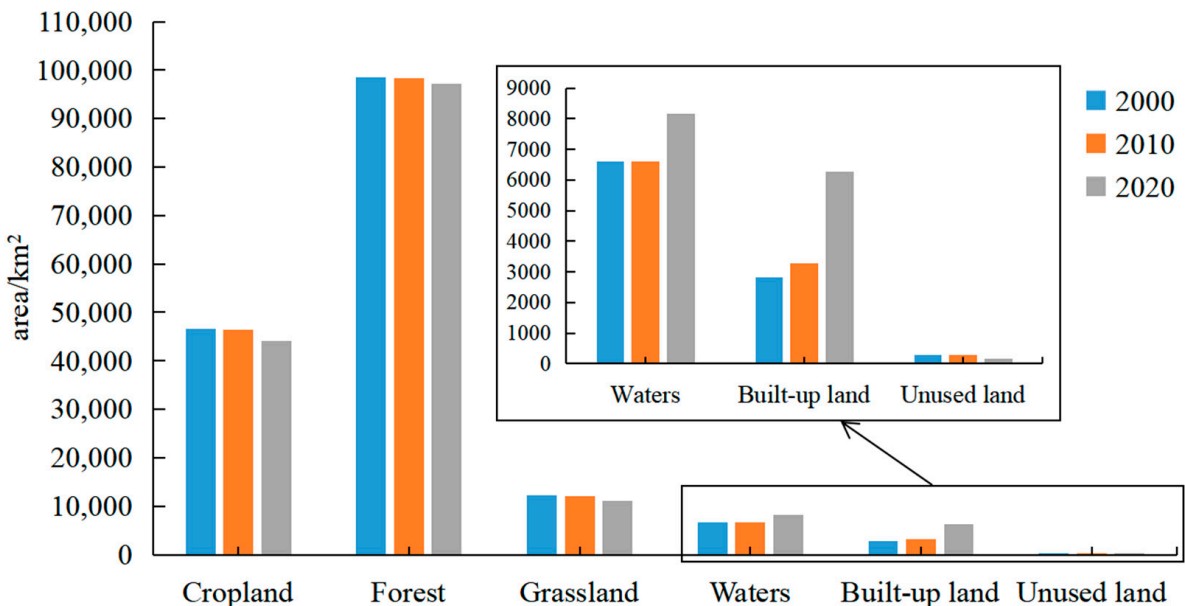

**Figure 4.** Land use area changes in Jiangxi Province, 2000–2020.

4.1.2. Land Use Transfer Analysis

Table 3 displays the land use transfer matrix for Jiangxi Province from 2000 to 2020. The highest contribution rates for both incoming and outgoing transfers were observed in forest, at 31.74% and 36.61%, respectively. This was followed by cropland, with contribution rates of 24.86% and 34.56%, respectively. As depicted in Table 3, the maximum area conversion occurred between cropland and forest, exceeding 4000 km$^2$.

**Table 3.** Land use transfer matrix, 2000–2020 (area—km$^2$; rate—%).

| Land Use Type | Cropland | Forest | Grassland | Waters | Built-Up Land | Unused Land | Total Transfers Out | Rate-Out |
|---|---|---|---|---|---|---|---|---|
| Cropland | 37,583.00 | 4556.03 | 1084.55 | 973.22 | 2483.73 | 13.20 | 9110.72 | 34.56 |
| Forest | 4168.60 | 88,791.57 | 3924.93 | 739.59 | 775.82 | 41.94 | 9650.89 | 36.61 |
| Grassland | 1604.67 | 3556.12 | 6011.68 | 392.77 | 618.96 | 18.98 | 6191.51 | 23.49 |
| Waters | 373.01 | 167.46 | 61.38 | 5909.28 | 78.10 | 20.46 | 700.41 | 2.66 |
| Built-up land | 381.14 | 52.02 | 28.20 | 32.26 | 2318.91 | 0.55 | 494.16 | 1.87 |
| Unused land | 25.71 | 35.33 | 28.20 | 112.94 | 9.65 | 67.48 | 211.83 | 0.80 |
| Total transfers in | 6553.13 | 8366.96 | 5127.26 | 2250.78 | 3966.27 | 95.13 | 26,359.52 | - |
| Rate-in | 24.86 | 31.74 | 19.45 | 8.54 | 15.05 | 0.36 | - | 100 |

Rate-in and rate-out stands for "transfer in contribution rate" and "transfer out contribution rate", respectively.

The expansion of built-up land was primarily sourced from cropland, forest, and grassland, accounting for areas of 2483.73 km$^2$, 775.82 km$^2$, and 618.96 km$^2$, respectively. Additionally, some built-up land was repurposed into other land use types, predominantly cropland and forest. And there has been significant conversion between waters, cropland, and forest. As the population grew, the rising demand for cropland and built-up land led to considerable changes in land use due to human activities. Consequently, forest, waters, and unused land were reclaimed for cultivation, while cropland and forest were repurposed into built-up land. In some cases, cropland was transformed into forest, and certain rural housing areas were reclaimed as cropland or forest, reflecting the impacts of population shifts and urbanization. Moreover, ecological lands—such as cropland, forest, grassland, and waters—have been protected under the provisions of national policy. Despite these changes, the area of unused land remains the smallest and has undergone minimal change.

### 4.2. Carbon Storage Change from 2000 to 2020

As depicted in Figure 5, the lowest value of carbon storage was observed in Poyang Lake, with the carbon storage gradually escalating in the surrounding areas. The carbon storage was relatively modest in the central and western basin and in the southeast. It was apparent that there was a significant relationship between carbon storage and topography. Flat areas are conducive to farming, production, and construction activities, while hills with higher topography are predominantly covered by trees, which are naturally occurring or planted, as they are difficult to develop. Consequently, regions with high topography exhibit higher carbon storage due to the forest's superior carbon density.

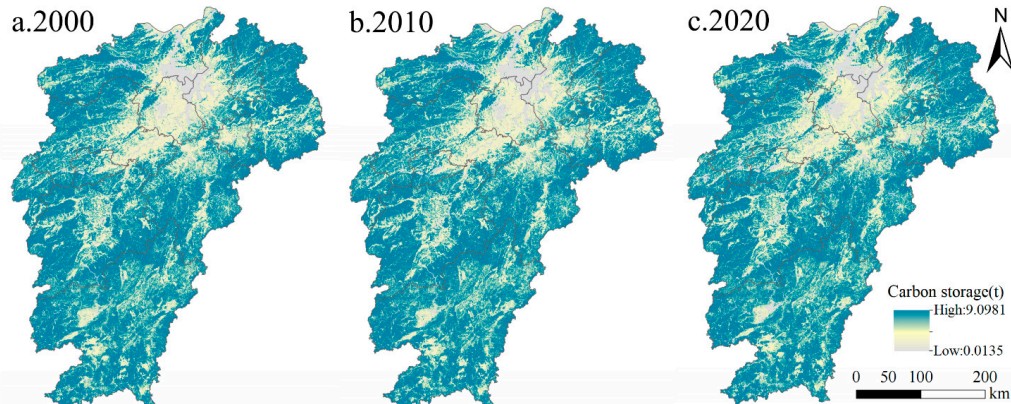

**Figure 5.** Spatial distribution of carbon storage in Jiangxi Province, 2000–2020.

As demonstrated in Figure 6, the forest's carbon storage was the predominant contributor to Jiangxi Province's carbon storage. This was followed by the carbon storage of cropland and grassland, while the carbon storage in waters, built-up land, and unused land were comparatively negligible. The carbon storage levels of Jiangxi Province amounted to $125,189.56 \times 10^4$ t, $124,959.29 \times 10^4$ t, and $122,306.57 \times 10^4$ t in 2000, 2010, and 2020, respectively, which means the carbon reserves decreased by $2882.99 \times 10^4$ t from 2000 to 2020, with $2652.72 \times 10^4$ t of this decrease occurring from 2010 to 2020—a contribution of 92.01%. The carbon reserves of the forest experienced the most significant decrease—$1297.93 \times 10^4$ t—followed by cropland and grassland, which decreased by $985.18 \times 10^4$ t and $664.3 \times 10^4$ t, respectively. Meanwhile, the carbon storage increased by $2.33 \times 10^4$ t and $65.27 \times 10^4$ t due to the expansion of area, despite the low carbon density of water and built-up land. Conversely, the carbon storage of unused land decreased by $3.19 \times 10^4$ t due to area reduction.

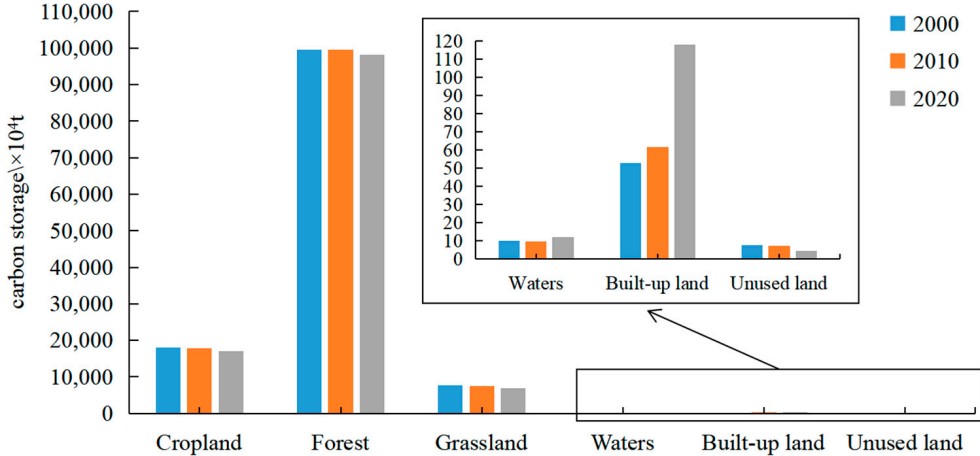

**Figure 6.** Carbon storage changes in Jiangxi Province, 2000–2020.

*4.3. Scenario Simulation in 2040*

4.3.1. Analysis of Driving Factors of Land Use Expansion

Land use changes are shaped by both natural and societal factors. For example, areas with flat terrain and a favorable climate are prime for production and construction activities, making them likely candidates for cropland and built-up land. Regions with a robust economy and convenient transportation are more prone to population clustering and subsequent expansion of built-up land [42]. Nevertheless, mountainous regions pose challenges for transformation into other land types due to their unsuitability for production activities. Therefore, we selected 13 driving factors to simulate the land use distribution in Jiangxi Province in 2040 (Table 1), and the contribution of each driving factor to land use change, as identified by the LEAS model of PLUS, is depicted in Figure 7.

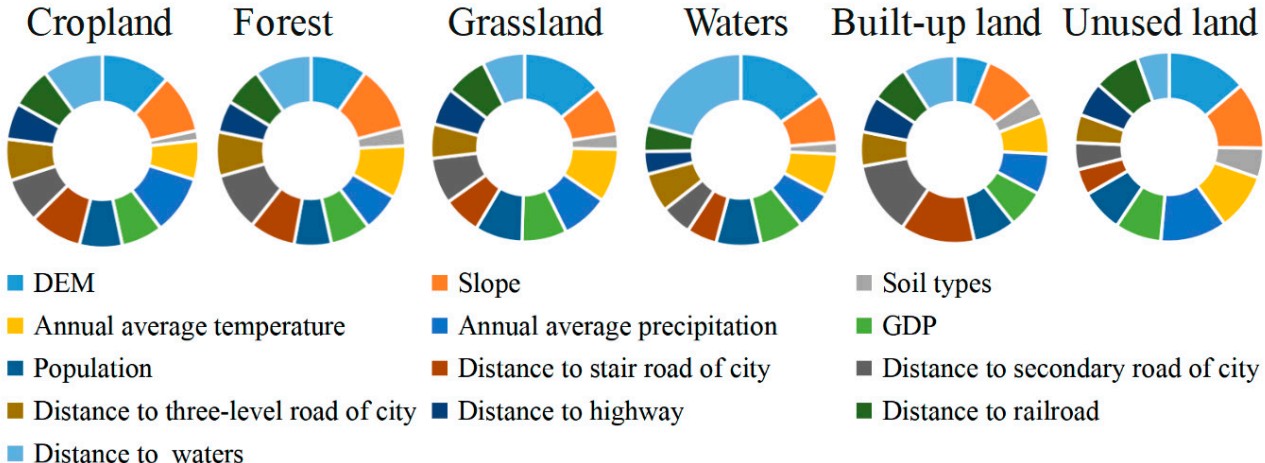

**Figure 7.** Distribution of contribution of driving factors.

The Digital Elevation Model (DEM) exerted the most considerable impact on the expansion of cropland, grassland, and unused land, contributing 0.117, 0.142, and 0.138, respectively. Among the 13 driving factors, slope had the highest contribution to forest expansion at 0.113. Meanwhile, the most significant factor influencing the growth of waters was the distance to the water system. It indicated that natural factors had a substantial impact on land use. As for social factors, the distance to the stair road, the second road of city, and the distance to railroad had the most significant impact on the expansion of cropland, forest, and unused land, respectively, being 0.088, 0.099, and 0.080, respectively. Population had had the greatest influence on the expansion of grassland and waters among social factors, which being 0.082 and 0.075, respectively. As for the influence on expansion of built-up land, the distance to the stair road of the city ranked first among all influencing factors at 0.128.

4.3.2. Land Use Scenario Simulation in 2040

The land use patterns and area changes for the S1, S2, S3, and S4 in 2040 were simulated by the CARS model of PLUS, as depicted in Figure 8 and Table 4.

Under S1, the area of cropland, forest, and grassland is projected to decline continuously by 2040, while the area of waters is expected to expand and built-up land to proliferate rapidly. As illustrated in Figure 8, the built-up land will primarily extend from urban areas, concentrating in the northern part of Jiangxi Province. As shown in Table 4, compared to 2020, the cropland area will increase by 2600.15 km² in S2, and the forest area will increase by 1342.96 km² in S3, with the area of cropland and forest increasing by 207.26 km² and 1503.12 km², respectively, in S4. However, we found that the protection of only one type of ecological land may lead to a greater loss of another type of ecological land. With the implementation of stricter protection measures, the available area for built-up land expansion will be gradually diminishing. It is worth mentioning that the area of built-up

land is projected to decrease by 255.34 km$^2$ under S4. Moreover, across the four scenarios, the area of waters will expand, while the area of unused land will decrease slightly.

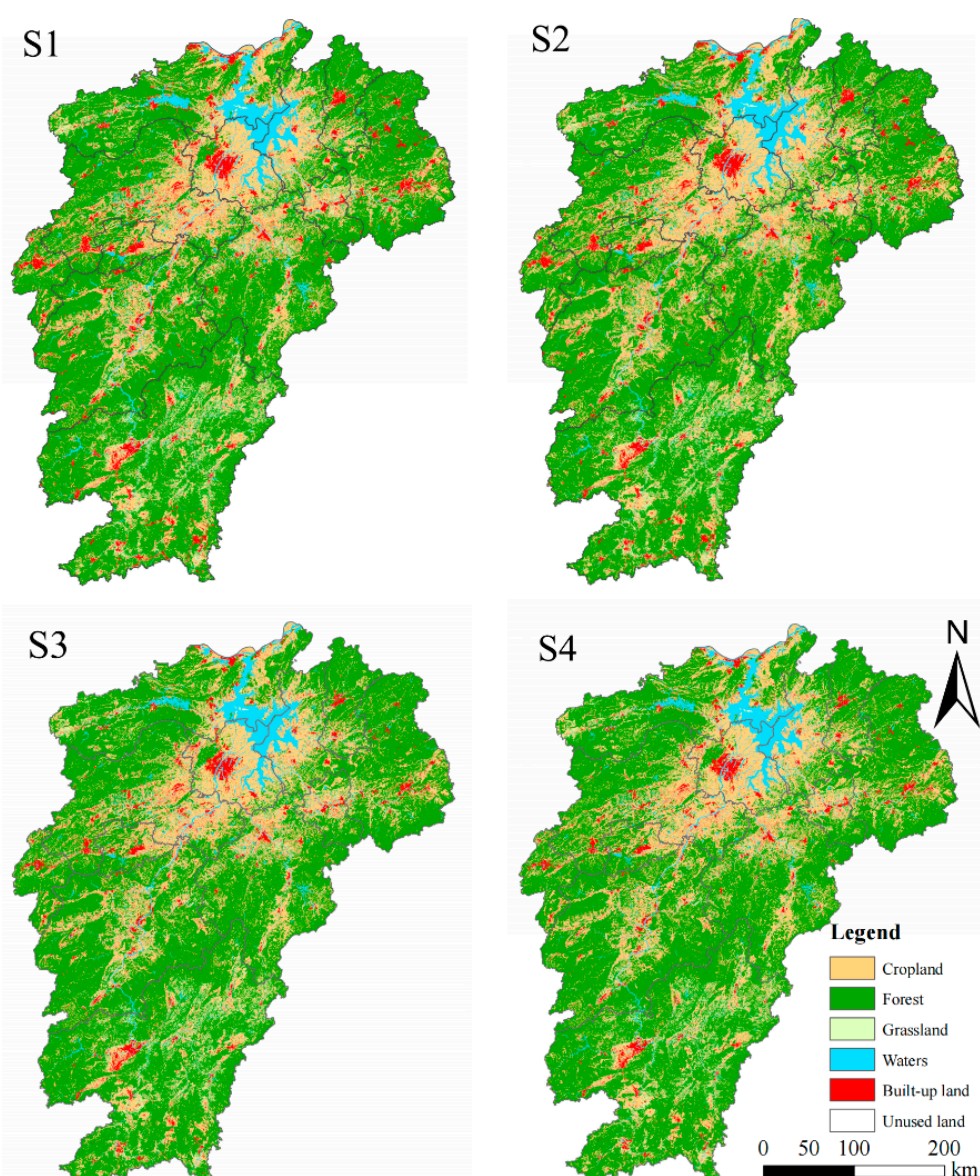

**Figure 8.** Spatial distribution of land use scenario simulation in 2040.

**Table 4.** Land use scenario simulation and area changes.

| Year/Period | Scenario | Cropland | Forest | Grassland | Waters | Built-Up Land | Unused Land |
|---|---|---|---|---|---|---|---|
| 2040 | S1 | 42,430.44 | 95,529.51 | 10,541.47 | 9441.57 | 8961.49 | 136.95 |
| | S2 | 46,736.28 | 93,376.27 | 10,028.90 | 8981.61 | 7787.65 | 130.72 |
| | S3 | 43,022.25 | 98,501.49 | 9807.25 | 9202.82 | 6370.76 | 136.86 |
| | S4 | 44,343.39 | 98,661.65 | 8943.58 | 8932.57 | 6029.84 | 130.41 |
| 2020–2040 | S1 | −1705.69 | −1629.02 | −597.47 | 1281.50 | 2676.31 | −25.65 |
| | S2 | 2600.15 | −3782.26 | −1110.04 | 821.54 | 1502.47 | −31.88 |
| | S3 | −1113.88 | 1342.96 | −1331.69 | 1042.75 | 85.58 | −25.74 |
| | S4 | 207.26 | 1503.12 | −2195.36 | 772.50 | −255.34 | −32.19 |

### 4.3.3. Carbon Storage Scenarios Simulation in 2040

Based on the simulations for the four scenarios, the integrated PLUS-InVEST model was employed to determine the distribution and changes in carbon storage within Jiangxi Province by 2040, as illustrated in Figure 9 and Table 5.

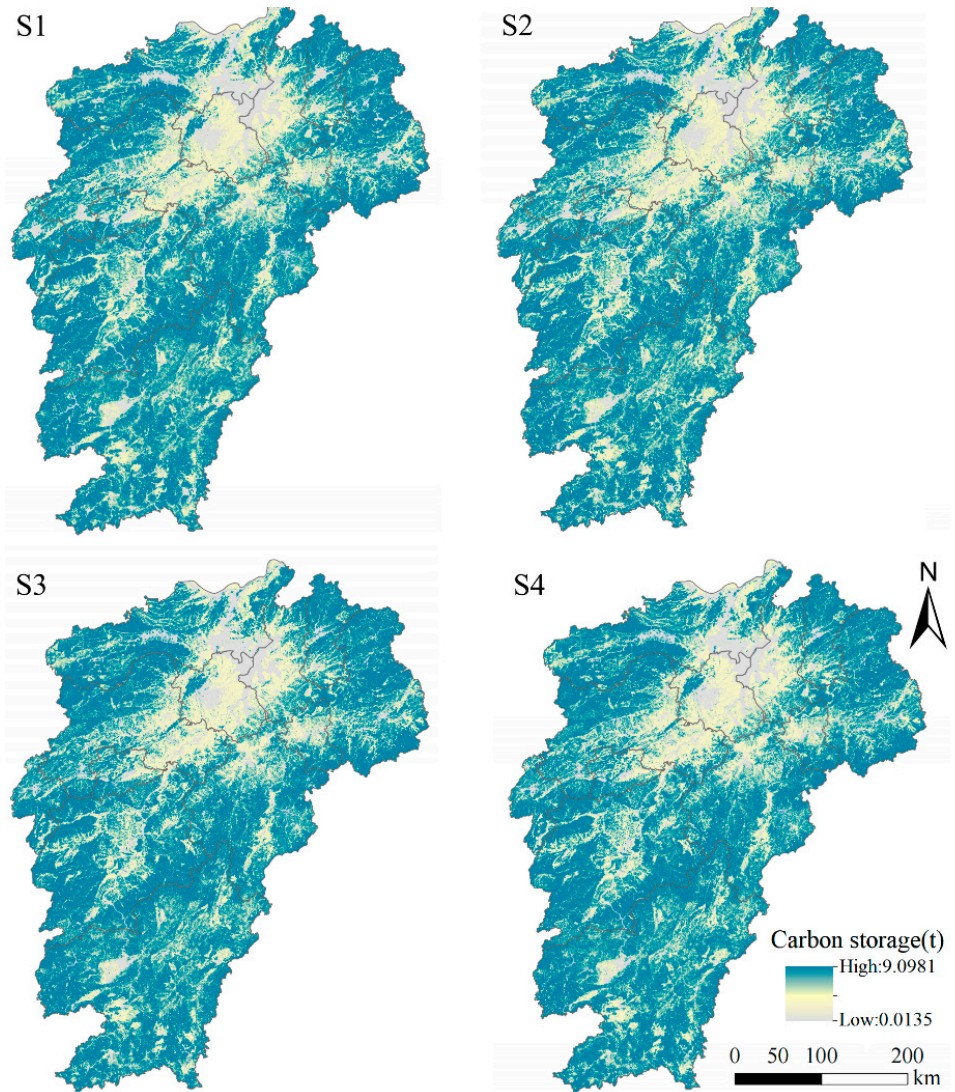

**Figure 9.** Spatial distribution of carbon storage scenarios simulation in 2040.

**Table 5.** Land use and carbon storage projections for Jiangxi Province in 2040 ($\times 10^4$ t).

| Year | Scenario | Cropland | Forest | Grassland | Waters | Built-Up Land | Unused Land | Total Carbon Storage |
|---|---|---|---|---|---|---|---|---|
| 2040 | S1 | 16,344.21 | 96,570.79 | 6579.99 | 14.16 | 168.48 | 3.74 | 119,681.36 |
| | S2 | 18,002.82 | 94,394.08 | 6260.04 | 13.47 | 146.41 | 3.57 | 118,820.39 |
| | S3 | 16,572.17 | 99,575.16 | 6121.69 | 13.8 | 119.77 | 3.74 | 122,406.33 |
| | S4 | 17,081.07 | 99,737.06 | 5582.58 | 13.4 | 113.36 | 3.56 | 122,531.04 |
| 2020–2040 | S1 | −657.03 | −1646.77 | −372.94 | 1.92 | 50.32 | −0.70 | −2625.21 |
| | S2 | 1001.58 | −3823.48 | −692.89 | 1.23 | 28.25 | −0.87 | −3486.18 |
| | S3 | −429.07 | 1357.60 | −831.24 | 1.56 | 1.61 | −0.70 | 99.76 |
| | S4 | 79.83 | 1519.50 | −1370.35 | 1.16 | −4.80 | −0.88 | 224.47 |

As shown in Figure 9, there will be no significant change in the distribution characteristics of carbon storage. However, it is clear that the carbon storage in urban expansion areas has evidently decreased, especially in Nanchang city. Specifically, by 2040, the carbon storage levels in S1, S2, S3, and S4 are projected to be $11,9681.36 \times 10^4$ t, $118,820.39 \times 10^4$ t, $122,406.33 \times 10^4$ t, and $122,531.04 \times 10^4$ t, respectively.

In S1, the carbon storage of cropland, forest, and grassland will all decrease, with the carbon storage of forest experiencing the greatest decline, amounting to $1646.77 \times 10^4$ t. The lowest carbon storage scenario occurs in S2, primarily due to a drastic reduction in forest carbon storage by $3823.48 \times 10^4$ t. S3 projects a rise of $99.76 \times 10^4$ t from 2020, primarily attributed to a $1357.6 \times 10^4$ t increase in forest carbon, while the carbon storage of cropland and grassland will decrease by $429.07 \times 10^4$ t and $831.24 \times 10^4$ t, respectively. S4 predicted the highest total carbon storage among the four scenarios, as a result of an increase in the carbon storage of cropland and forest by $79.83 \times 10^4$ t and $1519.5 \times 10^4$ t, respectively, leading to an overall carbon storage increase by $224.47 \times 10^4$ t compared with 2020. Additionally, under all four scenarios, the carbon storage of waters exhibits an increasing trend, while that of unused land displays a decreasing trend. However, these changes will be minimal. Except for S4, the carbon storage of built-up land is projected to see a slight increase by 2040. The results underscore the importance of ecological conservation in augmenting carbon storage. In terms of enhancing carbon storage, forest protection emerges as the pivotal strategy.

## 5. Discussion

### 5.1. Analysis of Changes in Land Use and Carbon Storage

It is evident that land use in Jiangxi Province underwent significant changes from 2000 to 2020, amidst rapid urbanization, population growth, and shifts in the natural environment. The study reveals that Jiangxi Province has experienced large-scale conversions between forest and cropland, as well as forest and grassland, and these were characterized by a decrease in the area of cropland, forest, grassland, and unused land, with an increase in the area of waters and built-up land. And the carbon storage demonstrated a declining trend due to land use change during this period. One the one hand, in order to protect and improve the ecological environment, China has implemented the policy of returning cropland to forest, which has led to the conversion from cropland to forest; on the other hand, the demand for cropland increased with population growth, and the original abandoned cropland and some shrubland that grew on low hills and gentle slopes have been transformed into cropland after land rearrangement, which led to the conversion from forest to cropland. And the reduction of forest was the main reason for the decline in carbon storage.

Moreover, it is worth noting that Jiangxi Province has been committed to developing specialized economic forest products in recent years, which transformed low-yielding forest into economic forests, such as oil tea, navel oranges, pomelo, and so on. And the Gannan navel orange has become a well-known product in China. During the early stages of forest reconstruction, the spacing between the newly planted trees was large, resulting in more bare ground. This initially led to a decline in forest coverage rate. However, with the increase in vegetation on the forest surface in abundance over time, the regional vegetation coverage will steadily improve.

The built-up land in Jiangxi Province has expanded rapidly over the past two decades, especially in 2010–2020. The expansion of built-up land was mainly to encroach upon substantial ecological areas, such as cropland and forest, which led to a decline in carbon storage. It has been consistently established that the rapid expansion of built-up land and the extensive conversion of cropland, forest, and grassland in recent years contribute significantly to carbon storage reduction and the greenhouse effect [43]. Protecting forest and appropriately controlling the expansion of built-up land have become necessary ways to enhance carbon storage.

### 5.2. Carbon Storage Development Strategy of Jiangxi Province

According to this study, under a natural development scenario, carbon storage is projected to further decrease in the future [44]. Once ecological land is converted to built-up land, it is challenging to restore the initial carbon storage level [45]. Therefore, implementing measures to effectively control the expansion of built-up land is essential. Jiangxi Province should adhere to the strict land use management system to protect the ecology, persist in adopting sensible ecological policies, and incorporate ecological principles into economic development planning. In January 2020, the Department of Natural Resources of Jiangxi Province promulgated the "Trial Measures for Natural Ecological Space Use Control in Jiangxi Province". This policy mandates maintaining a balance between ecological, agricultural, and construction spaces, strictly enforcing the ecological protection red line and permitting prudent development and structural adjustment of the general ecological space without compromising ecological functions or damaging ecosystems. This provides a robust framework for ecological protection.

Studies have shown that ecological restoration projects [7] and appropriate management measures [4,46,47] can effectively enhance carbon storage levels. With optimal land management practices, the amount of carbon storage that the world's terrestrial vegetation absorb can increase 13.74 Pg C annually [47]. Notably, forest has the highest carbon sequestration capacity, and Jiangxi Province's high forest coverage rate presents a valuable advantage in terms of carbon storage. To fully harness this potential, it is crucial for Jiangxi Province to prioritize the protection and sustainable management of forest resources. Jiangxi Province was one of the pioneering provinces to implement collective forest tenure reform of China in 2003, aiming to clarify forest property rights, reduce taxes and fees, liberalize operations, and regulate forest flow. The reform successfully enhanced forestry productivity and greatly boosted the enthusiasm of forest farmers for camping and afforestation [48,49]. However, as the reform entered its second decade in 2023, a recent survey on forest management has revealed that the low timber prices, poor access to forest, serious rural aging, and population outflow had contributed to the low motivation of foresters to manage their forest. These will be challenges for Jiangxi Province as it seeks to improve forestry reform and enhance the level of forest carbon storage in the future. It is crucial to progressively raise the ecological compensation standard for forestry [50] and augment financial support for forestry development [51]. It will not only help improve the motivation of foresters but also enable them to carry out effective forest management practices.

To address the reduction in vegetation cover caused by forest plantation renovation, it is important to focus on the development of understory economy in economic forests. This involves promoting the growth and utilization of understory plants, such as medicinal herbs, mushrooms, or other valuable non-timber forest products. By increasing the abundance of vegetation through the understory economy, not only can the ecological benefits be enhanced, but improved economic benefits for local communities can be produced. This approach promotes sustainable land use practices that balance both environmental and economic considerations.

Furthermore, Jiangxi Province should actively engage in the carbon storage market by establishing and improving policies and legal frameworks for trading carbon sinks. This will create a market-based mechanism to incentivize the sustainable development and management of carbon storage. By promoting the trading of carbon sinks, we can encourage the preservation and expansion of forests, which will contribute to the overall sustainability and resilience of the environment.

### 5.3. Uncertainty and Limitation

This article examines the process of land use and resulting changes in carbon storage in Jiangxi Province during 2000–2020 and presents four scenarios of future development. The study offers insightful information for future land use planning and ecological policy

formulation in the province. However, there are still uncertain factors and some limitations to this study.

Firstly, there are many more than 13 driving factors mentioned in this paper that impact land use, such as management systems, development philosophies, policies, and regulations, which were not comprehensively examined in this study. Secondly, the carbon density data used in the InVEST model were obtained from references, and the model ignored differences in vegetation types, growth conditions, and carbon density over time for the same land type. Therefore, while the InVEST model provides a useful estimate of changes in carbon storage caused by land use changes, its accuracy could be improved. Finally, the accuracy of land use raster data needs to be improved, and the remote sensing images may misidentify land use types. For example, remote sensing may identify economic forest industrial parks at the seedling stage as cropland, which may overestimate the conversion area of cropland and forest in this study.

## 6. Conclusions

Based on the coupled PLUS-InVEST model, this study assessed the changes of land use and carbon storage in Jiangxi Province from 2000 to 2020 and simulated four scenarios for 2040. The conclusions can be made are as follows:

(1) Land use in Jiangxi Province underwent significant changes in 2000–2020, which mainly occurred during 2010–2020. During 2000–2020, the area of cropland, forest, grassland, and unused land has declined, whereas the area of waters and built-up land has increased. Among the changes, the most significant decrease was observed in cropland, while the built-up land area experienced the most substantial increase. And there has been a large-scale conversion of cropland and forest.

(2) Land-use change resulted in a $2882.99 \times 10^4$ t reduction in carbon storage, with a decrease of 92.01% occurring from 2010 to 2020. Forests made the most significant contribution to the carbon storage of Jiangxi Province. By prioritizing the protection and management of forest resources, Jiangxi Province can play a significant role in mitigating climate change and ensuring a sustainable future.

(3) By 2040, under S1, areas of cropland, forest, and grassland are projected to decrease, while the area of cropland and forest will increase in S2 and S3, respectively, and the area of both cropland and forest will increase in S4. Moreover, the area of built-up land will keep growing except for S4, and the expansion area will be the largest under S1.

(4) By 2040, the carbon storage under S1, S2, S3, and S4 are projected to be $119{,}681.36 \times 10^4$ t, $118{,}820.39 \times 10^4$ t, $122{,}406.33 \times 10^4$ t, and $122{,}531.04 \times 10^4$ t, respectively. S4 is expected to yield the largest carbon storage while S2 is anticipated to result in the lowest. It is important to note that simply protecting cropland will not significantly increase carbon storage. To ensure efficient carbon storage in the future, it is crucial to maintain both cropland and ecological health.

**Author Contributions:** Conceptualization, Y.H., F.X. and Z.S.; methodology, Y.H., F.X. and Z.S.; data curation, Y.H.; supervision, F.X. and Z.S.; writing—original draft, Y.H.; writing—review and editing, Y.H., F.X. and S.Z.; visualization, F.X.; funding acquisition, S.Z. and Y.H. All authors have read and agreed to the published version of the manuscript.

**Funding:** This research was funded by the National Natural Science Foundation of China (Grant No. 71840013); the Jiangxi Selenium-rich Agriculture Special Project 2021 (Grant No. JXFXZD-2021-02); and the Graduate Student Innovation Program of School of Economics and Management, Jiangxi Agricultural University (Grant No. JG2022012).

**Data Availability Statement:** All the data used for the study appear in Section 2.2 of the article.

**Acknowledgments:** We value the reviewers' helpful recommendations.

**Conflicts of Interest:** The authors declare no conflict of interest.

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
