# Peer review of "Evolution and Multi-Scenario Prediction of Land Use and Carbon Storage in Jiangxi Province"

_forests, doi:10.3390/f14101933_

Round 1

Reviewer 1 Report

The manuscript aimed to study the changes in land use and their impact on carbon stocks in Jiangxi Province in China. The spatial patterns of land cover changes and carbon storage alterations from 2000 to 2020 and set four scenarios for 2040 were analyzed by using the coupled PLUS-InVEST model. However, the article lacked innovation, the author only analyzed the changes in the area of land types, but the analysis of driving factors of carbon storage changes caused by land type changes were insufficient. In addition, from the perspective of "Forests" magazine, readers are more concerned about how forest change leads to changes in carbon storage, and there is a serious lack of research on forests. Therefore, I think that the manuscript is not suitable for public publication.

1.Introduction.

Insufficient discussion on why the author used the coupled PLUS-InVEST model.

2.Study area and data.

(1)This paper classified land use types into six categories, unclear classification accuracy.

(2)Lack of driver factors analysis methods.

3. Discussion

What are the driving factors for changes in carbon storage that we hope to see? But there was not enough depth in the discussion.

 Minor editing of English language required

Reviewer 2 Report

There are some issues that need to be rectified, as flowing:

1.     Please provide a technical roadmap on the research methods in Section 3 to make the model, process, and logical order of the article clearer, improve the readability of the article, and enrich the visualization effect of the article.

2.     Why did the simulation analysis choose 2040? What's the reason? Do you have any special implications? Why not 30, 50, or 60? Please explain in 4.3.2. Land use scenario simulation in 2040.

3.     The same noun in the article has multiple descriptions that cause confusion and should be unified, such as:

a) cropland/arable land/ cultivated land/ agricultural land

b) Construction land/ built-up land

c) Forest/ Woodland

d) Carbon storage/ carbon stocks (These two terms have different academic meanings. Please choose the appropriate word according to the title of the article and unify).

4.     Lines 229 and 230: "4.1.2. Land Use Transfer Analysis Table 5 displays the land use transfer matrix for Jiangxi Province from 2000 to 2020. 230". The above description of Table 5 is an error; it should be “Table 3”. Please check and rectify.

5.     The format of Tables 3, 4, and 5 is not unified, with horizontal lines in the upper half and no horizontal lines in the lower half. Please unify the format.

6.     Lines 311 and 312 "As illustrated in Figure 6, the construction land will primarily extend from urban areas, concentrating in the northern part of Jiangxi Province." The above description of Figure 6 seems wrong (should be Figure 7); please check and rectify.

7.     Lines 331 and 332, "Specifically, by 2040, the carbon storage in S1, S2, S3, and S4 is projected to be 119681.36×104t, 118820.39×104t, 122406.33×104 t, and 122531.04×104 t, respectively." Above, the 104t should be 104 t (superscript required); please rectify. Meanwhile, check for the same error in the article and rectify it.

Minor editing of English language required
